# True prevalence of long-COVID in a nation-wide, population cohort study

Claire E. Hastie[1], David J. Lowe [1,2], Andrew McAuley[3,4], Nicholas L. Mills [5,6], Andrew J. Winter [7], Corri Black[8,9], Janet T. Scott[10], Catherine A. O'Donnell [1], David N. Blane [1], Susan Browne[1], Tracy R. Ibbotson[1] & Jill P. Pell [1] ✉

Long-COVID prevalence estimates vary widely and should take account of symptoms that would have occurred anyway. Here we determine the prevalence of symptoms attributable to SARS-CoV-2 infection, taking account of background rates and confounding, in a nationwide population cohort study of 198,096 Scottish adults. 98,666 (49.8%) had symptomatic laboratory-confirmed SARS-CoV-2 infections and 99,430 (50.2%) were age-, sex-, and socioeconomically-matched and never-infected. While 41,775 (64.5%) reported at least one symptom 6 months following SARS-CoV-2 infection, this was also true of 34,600 (50.8%) of those never-infected. The crude prevalence of one or more symptom attributable to SARS-CoV-2 infection was 13.8% (13.2%,14.3%), 12.8% (11.9%,13.6%), and 16.3% (14.4%,18.2%) at 6, 12, and 18 months respectively. Following adjustment for potential confounders, these figures were 6.6% (6.3%, 6.9%), 6.5% (6.0%, 6.9%) and 10.4% (9.1%, 11.6%) respectively. Long-COVID is characterised by a wide range of symptoms that, apart from altered taste and smell, are non-specific. Care should be taken in attributing symptoms to previous SARS-CoV-2 infection.

Estimates of the prevalence of long-COVID vary widely. The WHO estimates the percentage of people who continue to have, or develop, at least one symptom more than three months after SARS-CoV-2 infection as 10–20%[1]. However, the UK Office for National Statistics estimate that self-reported long-COVID at a population level is much lower at 2.7%[2]. By contrast, a recent meta-analysis of 194 studies including 735,006 participants estimated that, at an average follow-up of 126 days, 45% of COVID-19 survivors had at least one unresolved symptom[3].

Apart from altered taste and smell, the symptoms of long-COVID are non-specific. Symptoms attributed to long-COVID may be due to other causes, yet most studies of long-COVID do not include a comparison group. A study from the Netherlands compared persistent symptoms in 4231 participants who previously had COVID-19 and 8462 matched controls. Among the former, 21.3% of people had at least one symptom 3-5 months after SARS-CoV-2 infection compared with 8.7% of people not infected, suggesting the true prevalence may be nearer 12.6%[4].

The long-COVID in Scotland Study (Long-CISS) is a population cohort comprising people with laboratory-confirmed SARS-CoV-2 infection and an age-, sex-, and socioeconomically-matched group of people who have never been infected[5]. Using the Long-CISS cohort, we aimed to determine the true prevalence of long-COVID at 6, 12, and 18 months, overall and by sub-groups. This work expands on that previously published[5] by including additional waves of questionnaires and focusing analysis on the calculation of the prevalence of one or more ongoing symptom attributable to SARS-CoV-2 infection. Here, we show that the crude prevalence is 13.8%, 12.8%, and 16.3% at 6, 12,

[1]School of Health and Wellbeing, University of Glasgow, Glasgow, UK. [2]Emergency Department, Queen Elizabeth University Hospital, Glasgow, UK. [3]Public Health Scotland, Meridian Court, Glasgow, UK. [4]School of Health and Life Sciences, Glasgow Caledonian University, Glasgow, UK. [5]BHF Centre for Cardiovascular Science, University of Edinburgh, Edinburgh, UK. [6]Usher Institute, University of Edinburgh, Edinburgh, UK. [7]Sandyford Sexual Health Services, NHS Greater Glasgow and Clyde, Glasgow, UK. [8]Aberdeen Centre for Health Data Science, University of Aberdeen, Aberdeen, UK. [9]Public Health Directorate, NHS Grampian, Aberdeen, UK. [10]MRC-University of Glasgow Centre for Virus Research, University of Glasgow, Glasgow, UK. ✉e-mail: Jill.pell@glasgow.ac.uk

and 18 months respectively. Following adjustment for potential confounders, these figures are 6.6%, 6.5%, and 10.4% respectively.

## Results and discussion

Overall, 345,673 questionnaires were completed by 288,173 individuals, of whom 257,341 (89%) consented to record linkage to their test result. Following linkage, 53,530 were excluded because they reported a previous positive test that was not recorded on the database, and 5,715 because they had asymptomatic infections. Of the remaining 198,096 individuals, 98,666 (49.8%) had previous symptomatic, laboratory-confirmed SARS-CoV-2 infection and 99,430 (50.2%) had never had a positive test. PCR tests took place between the 20th of April 2020 and the 31st of May 2022. Questionnaires were completed between the 10th of May 2021 and the 14th of November 2022. Compared with those who did not provide consent, participants in the final sample were more likely to be female (58.8% vs 51.8%; *p*-value < 0.001), were older (>40 years 64.0% vs 51.1%; *p*-value < 0.001) and slightly more deprived (most deprived SIMD quintile 20.8% vs 20.4%; *p*-value < 0.001).

Infected individuals were less likely to have pre-existing health conditions and more likely to have been vaccinated (Table 1). Because new first infections occurred over time, later periods of the pandemic were less common in the never infected group. Whilst 64.5% reported at least one symptom six months following SARS-CoV-2 infection, this was also true of 50.8% of those never infected (Table 2). Results were similar at 12 (67.8% versus 55.0%) and 18 (72.6% versus 56.2%) months follow-up. The crude prevalence of at least one symptom attributable to SARS-CoV-2 infection was 13.8% (13.2%,14.3%), 12.8% (11.9%,13.6%), and 16.3% (14.4%,18.2%) at six, 12 and 18 months respectively. Following adjustment for potential confounders, these figures were 6.6% (6.3%, 6.9%), 6.5% (6.0%, 6.9%) and 10.4% (9.1%, 11.6%), respectively (Supplementary Table 1). The attributable prevalence was higher in women and those who had had more vaccination doses prior to infection and lower in those with more pre-existing health conditions (Fig. 1). The adjusted attributable percentage was higher for people infected later in the pandemic: 6.7% (6.2%, 7.1%) and 7.9% (6.9%, 9.0%) at six months follow-up for the delta and omicron variants respectively compared with 3.9% (3.2%, 4.6%) for the alpha variant (Supplementary Table 1).

Of the 98,666 participants with previous symptomatic infection 2256 (2.29%) had severe infection. At six months follow-up the crude prevalence of at least one symptom was 64.3% following mild infection compared with 79.3% following severe infection. These values were 67.8% and 82.5%, and 71.7% and 84.0%, respectively at 12 and 18 months follow-up.

Our finding that the true prevalence of long-COVID was 6.6–10.3% is not inconsistent with 12.7% reported in the Netherlands[4] and the WHO estimate of 10–20%[1]. Based on these three sources, the UK Office for National Statistics estimate of 2.7% may be an underestimate. In our previous analysis of the same cohort, 48% of people self-reported that they were not fully recovered six months following symptomatic SARS-CoV-2 infection[5]. Similarly, meta-analysis of published studies reported that 45% had unresolved symptoms at 4 months follow-up[3]. However, our findings from the current study suggest that whilst 64.5–72.6% of people report at least one symptom six to 18 months following SARS-CoV-2 infection, only 6.6%–10.3% are likely to have long-COVID. The symptoms of the remainder are likely to have occurred without SARS-CoV-2 infection but some people may mistakenly attribute them to long-COVID. Further work is required to refine the definition and diagnosis of long-COVID and support appropriate management.

A national cohort study in England used similar methodology to estimate long-COVID prevalence in adolescents aged 11–17 years[6]. Potential participants were invited from the individuals in Public Health England's SARS-CoV-2 testing database. Invitations to complete an online questionnaire were sent by letter, with a response rate of 13%. Those who tested positive for SARS-CoV-2 (*n* = 3065) were matched by

**Table 1 | Characteristics of participants by SARS-CoV-2 infection status**

| | Never infected N = 99,430 | Infected N = 98,666 | P value |
|---|---|---|---|
| | Med (IQR) | Med (IQR) | |
| Age (years) | 48 (33–59) | 46 (32-58) | <0.001 |
| **Sex** | N(%) | N(%) | |
| Female | 56,084 (56.4) | 60,429 (61.3) | <0.001 |
| Male | 43,346 (43.6) | 38,237 (38.8) | |
| **SIMD** | | | |
| 1 (most deprived) | 20,991 (21.1) | 20,134 (20.4) | <0.001 |
| 2 | 20,350 (20.5) | 19,907 (20.2) | |
| 3 | 18,673 (18.8) | 18,649 (18.9) | |
| 4 | 19,586 (19.7) | 19,565 (19.8) | |
| 5 (least deprived) | 19,830 (19.9) | 20,411 (20.7) | |
| **Ethnic group** | | | |
| White | 88,857 (89.4) | 91,546 (92.8) | <0.001 |
| South Asian | 1478 (1.49) | 1504 (1.52) | |
| Black | 607 (0.61) | 543 (0.55) | |
| Other | 2010 (2.02) | 1733 (1.76) | |
| Missing | 6478 (6.52) | 3340 (3.39) | |
| **Number of pre-existing health conditions** | | | |
| 0 | 67,491 (67.9) | 69,090 (70.0) | <0.001 |
| 1 | 13,822 (13.9) | 13,953 (14.1) | |
| 2-3 | 14,106 (14.2) | 12,673 (12.8) | |
| ≥4 | 4011 (4.03) | 2950 (2.99) | |
| **Pre-existing health conditions** | | | |
| Arthritis | 6942 (6.98) | 6143 (6.23) | <0.001 |
| Asthma/bronchitis/COPD | 22,991 (23.1) | 21,691 (22.0) | <0.001 |
| Cancer | 2057 (2.07) | 1404 (1.42) | <0.001 |
| CHD | 4044 (4.07) | 3224 (3.27) | <0.001 |
| Cystic fibrosis | 38 (0.04) | 31 (0.03) | 0.417 |
| Deep vein thrombosis | 401 (0.40) | 298 (0.30) | <0.001 |
| Depression/anxiety | 44,826 (45.1) | 42,244 (42.8) | <0.001 |
| Diabetes | 5252 (5.28) | 4799 (4.86) | <0.001 |
| High blood pressure | 9686 (9.74) | 9.114 (9.24) | <0.001 |
| HIV | 123 (0.12) | 114 (0.12) | 0.599 |
| Home oxygen | 65 (0.07) | 56 (0.06) | 0.438 |
| Kidney disease | 768 (0.77) | 673 (0.68) | 0.018 |
| Liver disease | 517 (0.52) | 338 (0.34) | <0.001 |
| Neurological condition | 2499 (2.51) | 1915 (1.94) | <0.001 |
| Overweight | 8914 (8.97) | 8149 (8.26) | <0.001 |
| Obese | 3278 (3.30) | 2688 (2.72) | <0.001 |
| Pulmonary embolism | 366 (0.37) | 274 (0.28) | <0.001 |
| Pulmonary fibrosis | 118 (0.12) | 78 (0.08) | 0.005 |
| Stroke | 971 (0.98) | 713 (0.72) | <0.001 |
| **Vaccinated** | | | |
| No | 64,022 (64.4) | 43,253 (43.8) | <0.001 |
| 1 dose | 5804 (5.84) | 6375 (6.46) | |
| 2 doses | 22,088 (22.2) | 27,974 (28.4) | |
| ≥3 doses | 7516 (7.56) | 21,064 (21.4) | |
| **Variant period** | | | |
| Pre VOC | 26,758 (26.9) | 16,309 (16.5) | <0.001 |
| No dominant (1) | 28,387 (28.6) | 15,786 (16.0) | |
| Alpha | 7269 (7.31) | 3796 (3.85) | |
| No dominant (2) | 3245 (3.26) | 2236 (2.27) | |
| Delta | 24,210 (24.4) | 29,061 (29.5) | |
| No dominant (3) | 3709 (3.73) | 9824 (9.96) | |
| Omicron | 5852 (5.89) | 21,654 (22.0) | |

*Med* median, *IQR* inter-quartile range, *N* number, *SIMD* Scottish Index of Multiple Deprivation, *COPD* chronic obstructive pulmonary disease, *CHD* coronary heart disease, *HIV* human immunodeficiency virus, *VOC* variant of concern.
Kruskal Wallis test for continuous variables, Chi2 test for categorical variables. All statistical tests are two-sided. P values are reported to three decimal places.

**Table 2 | Crude prevalence of individual and any symptoms at 6, 12 and 18 months following SARS-CoV-2 infection**

| | Infected N = 98,666 | | | Never infected N = 99,430 | | |
|---|---|---|---|---|---|---|
| | 6 months N = 64,733 N (%) | 12 months N = 24,338 N (%) | 18 months N = 6,538 N (%) | 6 months N = 68,146 N (%) | 12 months N = 26,493 N (%) | 18 months N = 3,974 N (%) |
| **Sensory** | | | | | | |
| Change in taste | 4364 (6.74) | 1944 (7.99) | 459 (7.02) | 953 (1.40) | 510 (1.93) | 84 (2.11) |
| Change in smell | 4984 (7.70) | 2286 (9.39) | 499 (7.63) | 735 (1.08) | 392 (1.48) | 59 (1.48) |
| Problems hearing | 3269 (5.05) | 1443 (5.93) | 451 (6.90) | 2154 (3.16) | 1062 (4.01) | 168 (4.23) |
| Problems with eyesight | 3516 (5.43) | 1598 (6.57) | 547 (8.37) | 2408 (3.53) | 1108 (4.18) | 204 (5.13) |
| Pins and needles | 5718 (8.83) | 2380 (9.78) | 855 (13.1) | 3961 (5.81) | 1709 (6.45) | 287 (7.22) |
| **Cardiorespiratory** | | | | | | |
| Chest pain | 3511 (5.42) | 1586 (6.52) | 490 (7.49) | 2031 (2.98) | 1035 (3.91) | 163 (4.10) |
| Palpitations | 4144 (6.40) | 1908 (7.84) | 651 (9.96) | 2149 (3.15) | 1064 (4.02) | 172 (4.33) |
| Breathlessness | 9803 (15.1) | 4465 (18.4) | 1560 (23.9) | 4877 (7.16) | 2530 (9.55) | 459 (11.55) |
| Dry cough | 8291 (12.8) | 3674 (15.1) | 1090 (16.7) | 4824 (7.08) | 2689 (10.15) | 412 (10.37) |
| Cough with phlegm | 7583 (11.7) | 3371 (13.9) | 954 (14.6) | 5933 (8.71) | 3326 (12.55) | 493 (12.41) |
| **Gastrointestinal** | | | | | | |
| Poor appetite | 3416 (5.28) | 1425 (5.86) | 414 (6.33) | 2991 (4.39) | 1394 (5.26) | 196 (4.93) |
| Abdominal pain | 4049 (6.25) | 1697 (6.97) | 534 (8.17) | 4042 (5.93) | 1802 (6.80) | 283 (7.12) |
| Sickness/vomiting | 3729 (5.76) | 1481 (6.09) | 493 (7.54) | 3431 (5.03) | 1532 (5.78) | 239 (6.01) |
| Diarrhea | 4862 (7.51) | 1957 (8.04) | 654 (10.0) | 4428 (6.50) | 1977 (7.46) | 310 (7.80) |
| Constipation | 2581 (3.99) | 1099 (4.52) | 360 (5.51) | 2530 (3.71) | 1136 (4.29) | 181 (4.55) |
| **Musculoskeletal** | | | | | | |
| Muscle aches/ weakness | 13,218 (20.4) | 5784 (23.8) | 1976 (30.2) | 9788 (14.4) | 4362 (16.5) | 735 (18.5) |
| Joint pain | 10,198 (15.8) | 4524 (18.6) | 1636 (25.0) | 9041 (13.3) | 4171 (15.7) | 744 (18.7) |
| **Neurological/mental health** | | | | | | |
| Headache | 14,869 (23.0) | 5880 (24.2) | 1723 (26.4) | 12,950 (19.0) | 5669 (21.4) | 779 (19.6) |
| Anxious/depressed | 10,781 (16.7) | 4383 (18.0) | 1381 (21.1) | 9015 (13.2) | 3667 (13.8) | 569 (14.3) |
| Confusion | 6182 (9.55) | 2720 (11.2) | 940 (14.4) | 3088 (4.53) | 1330 (5.02) | 234 (5.89) |
| Sleep problems | 13,521 (20.9) | 5905 (24.3) | 1780 (27.2) | 10,934 (16.0) | 4934 (18.6) | 721 (18.1) |
| Dizzy/blackouts/fits | 2659 (4.11) | 1104 (4.54) | 347 (5.31) | 1918 (2.81) | 863 (3.26) | 135 (3.40) |
| Balance problems | 3209 (4.96) | 1505 (6.18) | 512 (7.83) | 2246 (3.30) | 1070 (4.04) | 210 (5.28) |
| **Non-specific** | | | | | | |
| Tiredness | 25,992 (40.2) | 10,415 (42.8) | 3238 (49.5) | 19,633 (28.8) | 8274 (31.2) | 1287 (32.4) |
| Weight loss | 1165 (1.80) | 469 (1.93) | 138 (2.11) | 873 (1.28) | 380 (1.43) | 66 (1.66) |
| Skin rash | 2284 (3.53) | 909 (3.73) | 299 (4.57) | 1924 (2.82) | 773 (2.92) | 141 (3.55) |
| At least one symptom | 41,775 (64.5) | 16,507 (67.8) | 4743 (72.6) | 34,600 (50.8) | 14,581 (55.0) | 2234 (56.2) |

N number.

month of test, age, sex, and geographical region to adolescents who tested negative (*n* = 3739). At 3 months follow-up the crude prevalence of at least one symptom attributable to infection was 13.2%, very close to our estimate in adults of 13.7% at 6 months follow-up.

Symptoms, reduced quality of life, impairment of activities of daily living, and self-reported non or partial recovery following SARS-CoV-2 infection are more common among people with pre-existing health problems, especially multimorbidity[5]. However, the findings of this study did not support the conclusion that their worse health following SARS-CoV-2 infection is due to a higher prevalence of long-COVID. This is based on us applying a modification of the WHO definition of long-COVID as one or more persistent or new symptom. We could not examine whether, for example, their existing symptoms deteriorated more as a result of SARS-CoV-2 infection than would otherwise have occurred. Our modification of the WHO definition does not incorporate the minimum symptom duration of at least 2 months. We could not determine if the participant's reported symptoms lasted for at least this duration.

Whilst the percentage of people reporting one or more symptom at six months was slightly lower following omicron (63.3%) than the alpha (66.8%) and delta variants (66.7%), the true prevalence of long-COVID at six months was higher following omicron and delta than the alpha variant. Our adjusted result contradicts the findings of studies without comparison groups, that concluded that long-COVID is less prevalent following the omicron variant[7,8]. In a Norwegian prospective cohort study, Magnusson et al. found that, compared with individuals who tested negative for SARS-CoV-2, the risk of ongoing symptoms posed by the omicron and delta variants were comparable at 14-126 days follow-up[9].

Strengths of this study include its large, unselected study sample recruited from the general population, laboratory confirmation of infection status, and inclusion of a comparison group. To minimise bias, the comparison group was matched by age, sex and deprivation and we adjusted for a wide range of other confounders. Nonetheless, residual confounding is possible in any observational study and may explain the finding of a higher prevalence of long-COVID among people who had more vaccinations prior to infection. This finding conflicts with that of Antonelli et al.[10], who reported reduced odds of long-duration (≥28 days) symptoms following two vaccine doses compared with no vaccination.

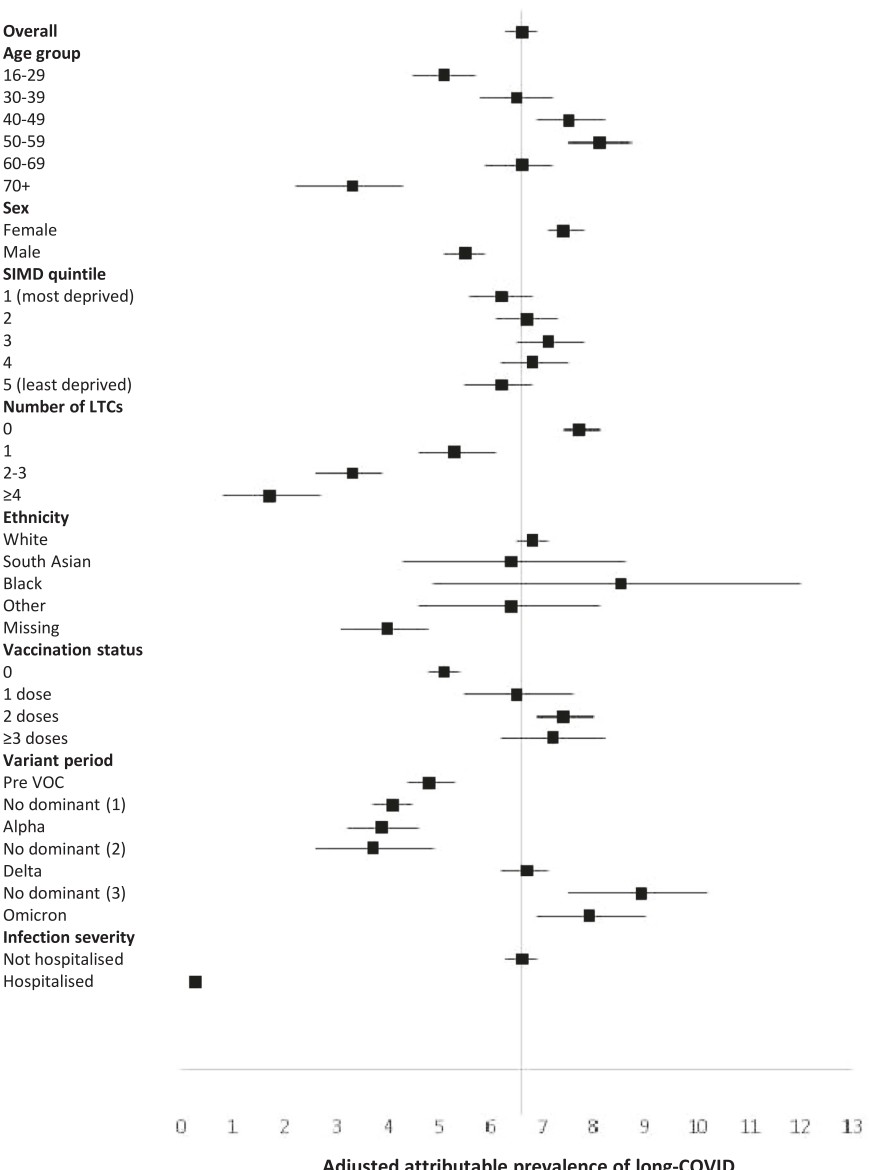

**Fig. 1 | Adjusted attributable prevalence of long-COVID at 6 months following symptomatic SARS-CoV-2 infection.** Data are presented as adjusted attributable prevalence values ±95% confidence intervals. SIMD Scottish Index of Multiple Deprivation; LTC Long term condition; VOC variant of concern. *N* = 132,879. Adjusted for age, sex, SIMD quintile, number of LTCs, ethnic group, vaccination status, and variant period. Numerical values of the estimates are provided in Supplementary Table 1.

Similarly, the apparent higher prevalence of long-COVID 18 months following infection may reflect the onset of new symptoms, residual confounding due to over-representation of infections early in the pandemic in spite of adjustment for dominant variants, or be due to retention bias whereby retention is higher in those with symptoms. Both groups gain 6 months of age between questionnaires and most symptoms increase with age. Moreover, both SARS-CoV-2 infection and many of the symptoms reported at follow-up vary by season. However, this is more likely to explain differences between 6 and 12 months follow-up and between 12 and 18 months. People dying from long-COVID over time could contribute to a fall in the prevalence of long-COVID over follow-up. However, our findings do not reflect such a fall.

Selection bias may be present in those who were tested for SARS-CoV-2, those who completed the questionnaire, and those who consented to linkage. During the time period when index PCR tests were conducted testing was available to everyone free of charge. However, people might be less likely to have been tested if their symptoms were mild resulting in some bias in testing. Furthermore, selection bias in questionnaire completion could potentially lead to overestimation of associations if having ongoing symptoms made participation more likely, or alternatively underestimation of associations if having more severe ongoing symptoms affected the ability to participate. In terms of linkage consent it is difficult to determine what direction of effect this might have. Despite this limitation our methodology represents a pragmatic recruitment method that allows representative response at a population level.

The crude prevalence of long-COVID was higher following severe infection than mild infection. However, we were unable to calculate adjusted attributable prevalence stratified by infection severity. Population attributable risk is not calculable by severity because it is a detailed version of the exposure variable (test status), meaning that severity and test status are strongly correlated. Future work should explore other indicators of severity and Covid-19 history.

There is the potential for misclassification bias. Antigen tests were not available. Moreover, some individuals in the comparison

group may have had SARS-CoV-2 infection that was not detected by a PCR test. This risk was reduced by excluding participants who had only negative PCR tests recorded but who self-reported that they had had SARS-CoV-2 infection. Nevertheless, the risk of classification error due to undiagnosed, asymptomatic infection remains.

## Methods

The Long-COVID in Scotland Study (Long-CISS) is an ambidirectional general population cohort. Every adult (>16 years) in Scotland with a positive PCR test was invited along with a comparison group who had had a negative test but never a positive test, matched by age, sex, deprivation quintile, and time period (in units of three-month periods)[5]. The latter were reallocated to the infected group if, and when, they tested positive. People who had asymptomatic SARS-CoV-2 infections were excluded. The National Health Service (NHS) Scotland platform that provided PCR result notifications identified eligible participants and invited them via automated SMS text messages. The COVID-19 & Respiratory Surveillance in Scotland Dashboard (https://scotland.shinyapps.io/phs-respiratory-covid-19/) provides information on testing and positivity rates over time. An online questionnaire (Supplementary Fig. 1), self-completed at six, 12 and 18 months following the index PCR test (first positive test or, for the comparison group, most recent negative test), collected information on pre-existing health conditions and 26 current symptoms (harmonised with the ISARIC questionnaire)[11].

Linkage to the test database provided the date and result of the index PCR test plus age, sex and postcode of residence. The latter was used to derive the Scottish Index of Multiple Deprivation (SIMD)[12]. Additional data were obtained through linkage to electronic health records - both five years prior their index test and subsequent to the test (up to January 2022) - on hospitalizations (Scottish Morbidity Record 01/04), dispensed prescriptions (Prescribing Information System), vaccinations, and death certificates (General Registrar Office). Severe infection was defined as hospital admission for SARS-CoV-2 infection. SARS-CoV-2 variants were defined as dominant if they accounted for ≥95% of cases genotyped that week in the UK population (https://sars2.cvr.gla.ac.uk/cog-uk/). Periods were defined as having no dominant variant when no single variant accounted for ≥95% of cases genotyped that week. Pre-existing health conditions were ascertained from self-report using the questionnaire, as well as linkage to previous hospitalizations and dispensed prescriptions. The methodology is described in detail elsewhere[5].

Our primary outcome was long-Covid, defined as one or more self-reported symptom at follow-up. Prevalence was calculated separately for those with previous symptomatic infection and those never infected. The crude attributable prevalence was estimated as the difference between these values. The adjusted attributable prevalence was calculated using the *regpar* command in Stata following logistic regression, adjusting for potential confounders (age, sex, deprivation quintile, ethnic group, individual and total number of long-term conditions, vaccination status, and dominant variant). Analysis was stratified by follow-up time; six, 12, and 18 months. Estimates were calculated for the whole study population, and then by subgroup.

### Ethics statement

Participants provided informed electronic consent for both data collection and data linkage, and study approval was obtained from the West of Scotland Research Ethics Committee (ref. 21/WS/0020) and the Public Benefit and Privacy Panel (ref. 2021-0180).

### Reporting summary

Further information on research design is available in the Nature Portfolio Reporting Summary linked to this article.

## Data availability

The datasets analysed during the current study (National Health Service Scotland's PCR testing platform, Scottish Morbidity Records 01 and 04, Prescribing Information System, Covid-19 vaccination database, and General Registrar Office death certificates) are available in the National Services Scotland National Safe Haven, https://www.isdscotland.org/Products-and-Services/eDRIS/Use-of-the-National-Safe-Haven/. This protects the confidentiality of the data and ensures that Information Governance is robust. Applications to access health data in Scotland are submitted to the NHS Scotland Public Benefit and Privacy Panel for Health and Social Care. Information can be found at https://www.informationgovernance.scot.nhs.uk/pbpphsc/.

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

## Acknowledgements

Long-CISS was funded by the Chief Scientist Office (ref. COV/LTE/20/06) and Public Health Scotland. We are grateful to Public Health Scotland, e-DRIS and HDR-UK for providing the routine data and access to the national safe haven, to Storm-ID for administering invitations and data

collection, the Scottish Government for supporting the study launch, and the University of Glasgow PPIE (Patient and Public Involvement and Engagement) groups for their contributions to study design, recruitment, and interpretation of results.

## Author contributions

J.P.P. had the original concept. J.P.P., D.J.L., A.J.W., C.E.H., C.A.O'D., D.N.B., N.L.M., C.B., J.T.S., T.R.I. and A.Mc.A. obtained funding. C.E.H., J.P.P., D.J.L. and A.Mc.A. obtained approvals. C.E.H. analysed the data. J.P.P., D.J.L., A.J.W., C.E.H., C.A.O'D., D.N.B., N.L.M., C.B., J.T.S., T.R.I., A.Mc.A. and S.B. interpreted the results. J.P.P. and C.E.H. produced the first draft. All authors revised the manuscript and approved the final version.

## Competing interests

The authors declare no competing interests.
