## [Peer Review File · Nature Communications]

True prevalence of long-COVID in a nationwide, population cohort studyREVIEWER COMMENTS

Reviewer #1 (Remarks to the Author):

Hastie and colleagues determined the 'true prevalence' of Long Covid in the general Scottish population using a largescale nationwide study. They found 6, 12, and 18 months following symptomatic infection 65%, 68% and 73% of people reported no, or incomplete, recovery respectively. Of note, the above symptoms were also commonly reported among test negative individuals at 6 (51%), 12 (55%) and 18 months (56%). This demonstrates the relative commonness of these symptoms in the population at any given time and subsequently the importance of a test-negative control group.

This ambidirectional, general population cohort study included people who had symptomatic laboratory-confirmed SARS-CoV-2 infections and a test negative comparison group, matched for age, sex, and deprivation, allowing novel insights into Long Covid that have not yet been extensively reported. On page 6, the authors make reference to their previous analysis of the same cohort. I think it would be helpful to the reader for the authors to provide clarity on how this work differs from and/or expands on that previously published in Nature Communications in October 2022.

The methods used to collect the data were valid, and the authors have provided a copy of self-reported symptom questionnaire. The manuscript is very well written; clear, precise, and easy to understand. The included figures and tables visually present complex data that is easy to interpret. The authors state whilst 64.5%-72.6% of people report at least one symptom 6-18 months following SARS-CoV-2 infection, only 6.6%-10.3% are likely to have long-COVID. They suggest the symptoms of the remainder are likely to have occurred without SARS-CoV-2 infection, but some people may mistakenly attribute them to long-COVID. Further explanation as to why this occurred and recommendations on how to avoid this misclassification during future research would be beneficial.

The authors conclude when taking into account the prevalence of Long Covid symptoms in non-infected matched controls and adjusting for potential confounders the prevalence of Long Covid is lower than previously reported in the literature (10.3% at 18 months). Though the authors make reference to the work by Ballering and colleagues I suggest they also include the work by Stephenson et al., 2022 ([https://www.thelancet.com/journals/lanchi/article/PIIS2352-4642\(22\)00022-0/fulltext](https://www.thelancet.com/journals/lanchi/article/PIIS2352-4642(22)00022-0/fulltext)), that also included a test negative control group. Since the authors state the attributable prevalence was higher in those who had received more vaccination doses prior to infection I encourage them to compare their findings to those by Antonelli and colleagues (<https://pubmed.ncbi.nlm.nih.gov/34480857/>).

Reviewer #2 (Remarks to the Author):

In the study "True prevalence of long-COVID in a nationwide, population cohort study" the authors aim to determine the prevalence of symptoms attributable to SARS-CoV-2 infection, taking account of background rates and confounding, in a nationwide population cohort study of 198,096 Scottish adults. The authors conclude that long covid symptoms cannot be linked to previous SARS-CoV-2 infection. The study has an interesting topic and results and could potentially add to the existing literature; however I have several concerns regarding the methods used and interpretation. Given many unclarities and limitations, I am not sure about the validity of findings and thus the significance to the field.

Major:

Selection into the study may greatly influence results. Several studies have shown that individuals testing negative, which make up the control group in this study, may be particularly health-conscious individuals who test themselves more often and also generally report more health complaints. Thus, the claim of no difference in prevalence for individuals testing positive vs

negative may be explained by selection bias. Please additionally include a control group of individuals who were non-tested. If not available, severe selection bias needs to be discussed including how it impacted on the estimates (from all sources, i.e. both from the choice of control group, from lack of questionnaire reporting and from non-consent to linkage).

Testing procedures in the community/study sample and how data on testing was managed in the study should be clearly described and their impact on results should be thoroughly discussed. Please provide data on how many tests were performed per individual and which test was selected in case of multiple tests. Most individuals have multiple negative and positive tests, and which test was chosen might impact on findings. Please describe and justify your approach.

For example, participants were included with an index PCR test and individuals testing negative were reallocated to the infected group if and when they tested positive. How was this handled in the study design and methods? If, for example, an individual was included in the negative group and reported their symptoms at six months, before testing positive at 8 months. Would he or she be in the positive group at 12 months? Wouldn't this method require a prospective design, which this study doesn't have? This is important to describe given that relatively few were infected early on during the pandemic and many were infected later.

Along this line, the design of the study is unclear. The authors name it a "population cohort study". Please describe under what circumstances this is a population cohort study. Is it prospective or retrospective? Why is this not a case control study – I think, from the descriptions, that cases were selected based on testing positive and controls testing negative were identified, retrospectively? Yet, it is not necessarily a case-control study, the methods including matching are too poorly described for the reader to judge. In the evidence hierarchy, case control studies using retrospective methods provide less strong evidence, for example on cause and effect, than prospective cohort studies. Prospective cohort studies on the topic do exist and should be referred to (example Magnusson et al., *Scientific Reports*, 2023). Given the design, the authors need to present descriptions of how individuals who could not continue their participation in the study were handled. Did they contribute in the study until they e.g. moved or died, and were censored from that time, or did you require complete followup? The latter would introduce further selection bias. If the authors think they have a prospective design, please use methods accordingly.

The matching needs a more thorough description. For example, time period was matched on, and from Table 1, it looks like the time unit included was based on dominant variant, typically lasting for half a year to a whole year. Time based biases might be introduced due to long durations of time matching units. Please describe and consider if time based biases can be minimized by design and methods. Could you match on calendar month or even week?

The aim of this study is not fully answered by the analyses performed. For example, Figure 1 only covers adjusted attributable prevalence at 6 months, not 12 or 18 months. And Table 1 includes many symptoms but is not broken down on subgroups. Please consider skipping one of these parts and perform a more thorough analyses of one of the sub-aims, i.e. focus either on all the symptoms and skip the subgroups, or focus on all the subgroups and provide one outcome.

The study needs a clear definition of what is meant by "long covid" and what are the clinical implications of the findings, given all the limitations particularly regarding selection bias and possible confounding. It is unclear what is being measured and of main interest in this study. "The outcome was one or more self-reported symptom at follow-up." However, the results section includes prevalence of a wide range of symptoms, which is inconsistent with methods and also inconsistent with the title. Do the authors measure long covid symptoms, or long covid?

The authors also need a better presentation of their data to support their conclusion and several claims in the discussion section. Confidence intervals of prevalence estimates are lacking, i.e. one cannot say that 64.5% is higher than 50.8% if CIs are overlapping which we do not know whether they are (and similar for the symptom reportings at 6, 12 months etc). Please provide CIs of all prevalence estimates, both crude and adjusted. For the adjusted estimates, please provide the prevalence with their CI, not only the adjusted difference with their CIs. This is important for the transparency of this study. It is always important to show the uncertainty around all estimates.

Minor:

From table 2, why do symptom reporting seem to increase from six months to 18 months? In previous research, complaints seem to decrease over time (see e.g. Skyrud et al., PlosOne, 2021). Again, confidence intervals are needed to determine if there is a real increase. Reasons for the potential increase, both for case and control group need to be discussed. Please consider running a within- and between individual model to take within-person clustering of symptoms over time into account.

The discussion section lacks important references and should include also what we know about prevalence from registerbased research. See for example omicron vs delta, Magnusson et al, Nat Comm 2022. Please also discuss the generalizability of findings and how self-reported data may be prone to bias and lack of generalizability. Along this line, selection bias is particularly important, for example that only 89% consented to linkage. And, very important, please provide data on the source population. How many were invited to respond to questionnaires and how many responded, at each measurement time during follow-up? What were the characteristics of non-responders and how might it impact on findings that these did not respond? As now, we only find the numbers and characteristics of responders yet we do not know what characterized non-responders or how many they were (participated at all visits during followup etc? Please see comment above on design).

It is controversial to include ethnicity in scientific research. Please focus on socioeconomy or residential region as markers of background.

Tables and figures should be self-explanatory. Please provide descriptions of abbreviations and measurements in legends.

Table 1 and Supplementary Table 1 contain massive amounts of data and is better presented in figure format, with confidence intervals.

I don't understand the data on variant, please describe pre VOC, no dominant 1, 2, etc

"Pre-existing health conditions were ascertained from self-report using the questionnaire, as well as linkage to previous hospitalizations and dispensed prescriptions." Please describe this linkage, how and when was it performed, what type of hospitalizations were included and for what period. How were the data compiled?

I can see the authors included the initial phase of the pandemic, from April 2020, when there was a lot of fear and anxiety. At the time, long covid was not known or established and symptom reporting may be increased just by the knowledge that one had the initial variant. Please include in discussion and consider omitting the first half year of testing in a sensitivity analyses. Also, to shed light on the generalizability of findings, please provide a timeline showing the number of tests from April 2020 to May 2022 as well as the proportion of all the tests being positive.

Why were people with asymptomatic infection excluded and how might it impact on findings? Again, please describe test criteria as one might wonder why asymptomatic individuals would be tested? Detail on methods with justification that are lacking: What were the criteria for being tested? Where and how did the testing take place? Was there routine testing, e.g. with hospital admission? How were home testing managed? Did the test criteria change during the inclusion period and were they equal for everyone included? Why were some previous positive tests not recorded on the database (and how might it impact on results)? Etc.

Are there data on severe covid-19, i.e. whether individuals were hospitalized at or around their test dates? Along this line, to whom do the findings apply? I think this needs to be made more clear in abstract, title etc.

Reviewer #3 (Remarks to the Author):

Thank you for the opportunity to review this paper. Examining the symptom burden of long COVID using a matched cohort framework provides us a clearer understanding of the symptoms attributable to long COVID. As the authors point out, this approach has been examined in other populations and the addition of this large cohort is useful to get closer to understanding the prevalence of this condition. However, I have a few suggestions below that could benefit the reader. Many of my questions are around how the symptom onset and duration was captured, which is important to clarify when applying the WHO definition, as new onset and at least 2 months of duration are key components.

Clarifying Questions

1. Ln 154: "PCR tests took place between the 20th of April 2020 and the 31st of May 2022. Questionnaires were completed between the 10th of May 2021 and the 14th of November 2022." Given these dates, it seems possible that for some participants, the initial questionnaire was administered at least one year since the index COVID infection. Would these participants have only contributed data at the 12 month timepoint? Please clarify.
2. Ln 159: Please clarify how you determined the variant of concern for the uninfected matched group. From Hastie 2022 Nat Comm paper it appears this was the variant of concern at the time of study invitation.
3. Ln 189: "This is based on us applying the WHO definition of long-COVID as one or more persistent or new symptom."
 - a. How were the symptoms for both infected and matched controls determined to be new onset? Was it from Q9 on the questionnaire? If Q9 was utilized to determine new onset, then please comment on the possibility that exacerbation of an existing symptom prior to COVID infection was captured since the participants were asked "Are any of these new or worse since your Covid-19 test?".
 - b. The WHO definition emphasizes the minimum symptom duration of at least 2 months. How did you determine if the participant's reported symptoms lasted for a least this duration? From my reading of the questionnaire, it does seem possible that the patient may report a new symptom since COVID infection but the duration may not have been at least 2 months.
4. Ln 213: "...with a positive PCR test was invited along with a comparison group who had had a negative test but never a positive test" Please comment on the availability of antigen tests for the cohort. Could someone with a positive antigen test be contained in the matched cohort group? Please comment on this potential misclassification if so.
5. Ln 217: "An SMS text prompted online questionnaire (Supplementary Figure 1), self-completed at six, 12 and 18 months following the index PCR test"
 - a. Could you comment on the response rate to the text questionnaire by infection status?
 - b. Can you describe further how SMS texts were sent out? From Hastie 2022 Nat Comm paper it appears the this was at health clinic level but also contact tracing databases.
6. Table 1: Can you further describe the time periods used for the variant periods, especially "no dominant"?

REVIEWER COMMENTS

Reviewer #1 (Remarks to the Author):

Hastie and colleagues determined the 'true prevalence' of Long Covid in the general Scottish population using a largescale nationwide study. They found 6, 12, and 18 months following symptomatic infection 65%, 68% and 73% of people reported no, or incomplete, recovery respectively. Of note, the above symptoms were also commonly reported among test negative individuals at 6 (51%), 12 (55%) and 18 months (56%). This demonstrates the relative commonness of these symptoms in the population at any given time and subsequently the importance of a test-negative control group.

This ambidirectional, general population cohort study included people who had symptomatic laboratory-confirmed SARS-CoV-2 infections and a test negative comparison group, matched for age, sex, and deprivation, allowing novel insights into Long Covid that have not yet been extensively reported. On page 6, the authors make reference to their previous analysis of the same cohort. I think it would be helpful to the reader for the authors to provide clarity on how this work differs from and/or expands on that previously published in Nature Communications in October 2022.

The following sentence has been added "This work expands on that previously published by including additional questionnaire waves and focusing analysis on the calculation of the prevalence of ongoing symptoms attributable to SARS-CoV-2 infection."

The methods used to collect the data were valid, and the authors have provided a copy of self-reported symptom questionnaire. The manuscript is very well written; clear, precise, and easy to understand. The included figures and tables visually present complex data that is easy to interpret. The authors state whilst 64.5%-72.6% of people report at least one symptom 6-18 months following SARS-CoV-2 infection, only 6.6%-10.3% are likely to have long-COVID. They suggest the symptoms of the remainder are likely to have occurred without SARS-CoV-2 infection, but some people may mistakenly attribute them to long-COVID. Further explanation as to why this occurred and recommendations on how to avoid this misclassification during future research would be beneficial.

The following sentence has been added "Further work is required to refine the definition and diagnosis of long-COVID and support appropriate management."

The authors conclude when taking into account the prevalence of Long Covid symptoms in non-infected matched controls and adjusting for potential confounders the prevalence of Long Covid is lower than previously reported in the literature (10.3% at 18 months). Though the authors make reference to the work by Ballering and colleagues I suggest they also include the work by Stephenson et al., 2022 ([https://www.thelancet.com/journals/lanchi/article/PIIS2352-4642\(22\)00022-0/fulltext](https://www.thelancet.com/journals/lanchi/article/PIIS2352-4642(22)00022-0/fulltext)), that also included a test negative control group. Since the authors state the attributable prevalence was higher in those who had had received more vaccination doses prior to infection I encourage them to compare their findings to those by Antonelli and colleagues (<https://pubmed.ncbi.nlm.nih.gov/34480857/>).

Stephenson et al. (2022) and Antonelli et al. (2022) have been added to the discussion.

Reviewer #2 (Remarks to the Author):

In the study “True prevalence of long-COVID in a nationwide, population cohort study” the authors aim to determine the prevalence of symptoms attributable to SARS-CoV-2 infection, taking account of background rates and confounding, in a nationwide population cohort study of 198,096 Scottish adults. The authors conclude that long covid symptoms cannot be linked to previous SARS-CoV-2 infection. The study has an interesting topic and results and could potentially add to the existing literature; however I have several concerns regarding the methods used and interpretation. Given many unclarities and limitations, I am not sure about the validity of findings and thus the significance to the field.

Major:

Selection into the study may greatly influence results. Several studies have shown that individuals testing negative, which make up the control group in this study, may be particularly health-conscious individuals who test themselves more often and also generally report more health complaints. Thus, the claim of no difference in prevalence for individuals testing positive vs negative may be explained by selection bias. Please additionally include a control group of individuals who were non-tested. If not available, severe selection bias needs to be discussed including how it impacted on the estimates (from all sources, i.e. both from the choice of control group, from lack of questionnaire reporting and from non-consent to linkage).

Potential participants were invited using contact details held by NHS Scotland in their PCR result database used for the SARS-CoV-2 test and protect initiative. Therefore, it is not possible to include a control group who were not tested.

People who did not get tested were also subject to potential bias and were more prone to undiagnosed SARS-CoV-2 infection.

The possibility of selection bias has been added to the discussion.

Testing procedures in the community/study sample and how data on testing was managed in the study should be clearly described and their impact on results should be thoroughly discussed. Please provide data on how many tests were performed per individual and which test was selected in case of multiple tests. Most individuals have multiple negative and positive tests, and which test was chosen might impact on findings. Please describe and justify your approach.

We do not know how many tests were performed per individual because we do not have access to these individual level data. The index test was the first positive test or, for the comparison group, most recent negative test. This information has been added to the methods.

For example, participants were included with an index PCR test and individuals testing negative were reallocated to the infected group if and when they tested positive. How was this handled in the study design and methods? If, for example, an individual was included in the negative group and reported their symptoms at six months, before testing positive at 8 months. Would he or she be in the positive group at 12 months? Wouldn't this method require a prospective design, which this study doesn't have? This is

important to describe given that relatively few were infected early on during the pandemic and many were infected later.

Yes, this individual would be in the positive group at 12 months with length of follow-up adjusted according to the date of their positive test (i.e. the clock restarts when they become positive). Their follow-up as an initial negative participant was censored at the time they become positive. Long-CISS employs an ambidirectional study design – that is, it includes both retrospective and prospective components. This information has been added to the methods.

Along this line, the design of the study is unclear. The authors name it a “population cohort study”. Please describe under what circumstances this is a population cohort study. Is it prospective or retrospective? Why is this not a case control study – I think, from the descriptions, that cases were selected based on testing positive and controls testing negative were identified, retrospectively? Yet, it is not necessarily a case-control study, the methods including matching are too poorly described for the reader to judge. In the evidence hierarchy, case control studies using retrospective methods provide less strong evidence, for example on cause and effect, than prospective cohort studies. Prospective cohort studies on the topic do exist and should be referred to (example Magnusson et al., Scientific Reports, 2023). Given the design, the authors need to present descriptions of how individuals who could not continue their participation in the study were handled. Did they contribute in the study until they e.g. moved or died, and were censored from that time, or did you require complete followup? The latter would introduce further selection bias. If the authors think they have a prospective design, please use methods accordingly.

The study is a population cohort study. The population is the Scottish general population from which participants were recruited. If they were recruited from e.g. hospital attendance, it would have been a hospital cohort. The study is an ambidirectional cohort study. An ambidirectional cohort combines both retrospective and prospective cohort designs. It recruits historical cases and adds new incident cases prospectively. It is not a case control study. A case-control study classifies and recruits people based on the outcomes of interest. Therefore, a case control study would recruit people with long-COVID (cases) and people without long-COVID (controls), then ascertain their exposure of interest. In contrast, a cohort study classifies and recruits people based on their exposure of interest. Therefore, it recruits people with SARS-CoV-2 infection (exposed) and people without SARS-CoV-2 infection (not exposed), then ascertains their outcome of interest. Cohort studies are a stronger epidemiological design largely because case-control studies are prone to survival bias. Both cohort and case-control studies provide evidence of association that should not be assumed to be causal. Temporality is one of the criteria that strengthens the likelihood of an association being causal. In our study the test date precedes the questionnaire date by at least 6 months. Analysis is based on completed questionnaires at follow-up. Therefore, people who do not complete questionnaires (due to death or other reason) are not included (in either the numerator or denominator). We have added a note that people dying from long-COVID over time could contribute to a fall in the prevalence of long-COVID over follow-up.

Reference to Magnusson et al. has been added to the manuscript.

The matching needs a more thorough description. For example, time period was matched on, and from Table 1, it looks like the time unit included was based on dominant variant, typically lasting for half a year to a whole year. Time based biases might be introduced due to long durations of time matching

units. Please describe and consider if time based biases can be minimized by design and methods. Could you match on calendar month or even week?

Three-month periods were used to group potential participants and match positive and negative tests. This information has been added to the methods. The time units based on dominant variant were only used to adjust multivariate analyses.

The aim of this study is not fully answered by the analyses performed. For example, Figure 1 only covers adjusted attributable prevalence at 6 months, not 12 or 18 months. And Table 1 includes many symptoms but is not broken down on subgroups. Please consider skipping one of these parts and perform a more thorough analyses of one of the sub-aims, i.e. focus either on all the symptoms and skip the subgroups, or focus on all the subgroups and provide one outcome.

Overall adjusted attributable prevalence estimates are provided for each follow-up time in the text. Figure 1 presents subgroup estimates at 6-month follow-up because this is the largest sample.

The study needs a clear definition of what is meant by “long covid” and what are the clinical implications of the findings, given all the limitations particularly regarding selection bias and possible confounding. It is unclear what is being measured and of main interest in this study. “The outcome was one or more self-reported symptom at follow-up.” However, the results section includes prevalence of a wide range of symptoms, which is inconsistent with methods and also inconsistent with the title. Do the authors measure long covid symptoms, or long covid?

Our primary outcome is long-Covid, defined as one or more self-reported symptom at follow-up. This is determined from the response to a long tick list of symptoms experienced within the last week. Table 2 is included to provide information on the individual symptoms reported and their prevalence.

The authors also need a better presentation of their data to support their conclusion and several claims in the discussion section. Confidence intervals of prevalence estimates are lacking, i.e. one cannot say that 64.5% is higher than 50.8% if CIs are overlapping which we do not know whether they are (and similar for the symptom reporting at 6, 12 months etc). Please provide CIs of all prevalence estimates, both crude and adjusted. For the adjusted estimates, please provide the prevalence with their CI, not only the adjusted difference with their CIs. This is important for the transparency of this study. It is always important to show the uncertainty around all estimates.

We have added confidence intervals around the crude attributable prevalences. Confidence intervals for the adjusted attributable prevalences were already included in the Figure but have now been added to the text. We have not added confidence intervals around the actual prevalences in either group as these are not estimates.

Minor:

From table 2, why do symptom reporting seem to increase from six months to 18 months? In previous research, complaints seem to decrease over time (see e.g. Skyrud et al., PlosOne, 2021). Again, confidence intervals are needed to determine if there is a real increase. Reasons for the potential

increase, both for case and control group need to be discussed. Please consider running a within- and between individual model to take within-person clustering of symptoms over time into account.

There are a lot of potential explanations for changes by stage of follow-up. In the infected group it may be due to a real increase in long-COVID due to re-emergence or new occurrence of symptoms or be due to retention bias whereby retention is higher in those with symptoms. Both groups gain 6 months of age between questionnaires and most symptoms increase with age. Both SARS-CoV-2 infection and many of the symptoms reported at follow-up vary by season. However, this is more likely to explain differences between 6 and 12 months follow-up and between 12 and 18 months (not between 6 and 18 months which would occur during the same season). We have added this to the limitations section. We have already reported between individual results. Reporting within individual changes would markedly reduce statistical power. We report confidence intervals for derived estimates (crude and adjusted attributable percentages) but not for absolute prevalence as per convention.

The discussion section lacks important references and should include also what we know about prevalence from registerbased research. See for example omicron vs delta, Magnusson et al, Nat Comm 2022. Please also discuss the generalizability of findings and how self-reported data may be prone to bias and lack of generalizability. Along this line, selection bias is particularly important, for example that only 89% consented to linkage. And, very important, please provide data on the source population. How many were invited to respond to questionnaires and how many responded, at each measurement time during follow-up? What were the characteristics of non-responders and how might it impact on findings that these did not respond? As now, we only find the numbers and characteristics of responders yet we do not know what characterized non-responders or how many they were (participated at all visits during followup etc? Please see comment above on design).

We have limited, aggregated data on non-responders. The following has been added to the text "Compared with those who did not provide consent, participants in the final sample were more likely to be female (58.8% vs 51.8%; p-value <0.001), were older (>40 years 64.0% vs 51.1%; p-value <0.001) and slightly more deprived (most deprived SIMD quintile 20.8% vs 20.4%; p-value <0.001)."

It is controversial to include ethnicity in scientific research. Please focus on socioeconomic or residential region as markers of background.

We disagree completely with this statement. Whilst ethnicity and socioeconomic status may be correlated to some extent, they are completely different concepts. We have data on both and have included both. Previous studies have provided consistent evidence of differences in infection rates and outcomes by ethnic group, independent of socioeconomic status. Therefore, to ignore this as a potential confounder would be inappropriate.

Tables and figures should be self-explanatory. Please provide descriptions of abbreviations and measurements in legends.

Where missing, this information has been added to the manuscript.

Table 1 and Supplementary Table 1 contain massive amounts of data and is better presented in figure format, with confidence intervals.

We disagree. Table 1 is summary data and, as per convention, is presented as a Table. We believe both would be messy, bigger and harder to interpret as Figures.

I don't understand the data on variant, please describe pre VOC, no dominant 1, 2, etc

The variant of concern is defined as the dominant variant ($\geq 95\%$ of infections) in the UK at the time of index test. Periods were defined as having no dominant variant when no single variant accounted for $\geq 95\%$ of cases genotyped that week. This has been added to the methods, as has the Cog-UK Mutation Explorer website. Pre-VOC is the period early in the pandemic prior to any new, dominant variants of the original pathogen being identified.

“Pre-existing health conditions were ascertained from self-report using the questionnaire, as well as linkage to previous hospitalizations and dispensed prescriptions.” Please describe this linkage, how and when was it performed, what type of hospitalizations were included and for what period. How were the data compiled?

The following has been added to the methods “Additional data were obtained through linkage to electronic health records - both five years prior to their index test and subsequent to the test (up to January 2022) - on hospitalizations (Scottish Morbidity Record 01/04), dispensed prescriptions (Prescribing Information System), vaccinations, and death certificates (General Registrar Office).”

I can see the authors included the initial phase of the pandemic, from April 2020, when there was a lot of fear and anxiety. At the time, long covid was not known or established and symptom reporting may be increased just by the knowledge that one had the initial variant. Please include in discussion and consider omitting the first half year of testing in a sensitivity analyses. Also, to shed light on the generalizability of findings, please provide a timeline showing the number of tests from April 2020 to May 2022 as well as the proportion of all the tests being positive.

We did not ask people if they have long-COVID. We asked if they have any of a long list of symptoms. We then defined long-COVID based on their responses. We have already included in the analysis the phase of the pandemic (using VOC data). Omitting people infected early in the pandemic would be inappropriate as we would lose the people with the longest follow-up and the results would not be representative of the pandemic as a whole.

The COVID-19 & Respiratory Surveillance in Scotland Dashboard (<https://scotland.shinyapps.io/phs-respiratory-covid-19/>) provides information on testing and positivity rates over time. This has been added to the methods section.

Why were people with asymptomatic infection excluded and how might it impact on findings? Again, please describe test criteria as one might wonder why asymptomatic individuals would be tested? Detail on methods with justification that are lacking: What were the criteria for being tested? Where and how did the testing take place? Was there routine testing, e.g. with hospital admission? How were home testing managed? Did the test criteria change during the inclusion period and were they equal for

everyone included? Why were some previous positive tests not recorded on the database (and how might it impact on results)? Etc.

PCR testing was freely available to the population for most of the period covering index tests. The reason for the test being conducted is not recorded. However, during the pandemic certain occupational groups (hospital and social care workers) were routinely tested as were people travelling between countries. This resulted in the detection of asymptomatic infections. Our previous publication (Hastie, C.E., et al. Outcomes among confirmed cases and a matched comparison group in the Long-COVID in Scotland study. Nature Communications 13, 5663 (2022)) found that asymptomatic SARS-CoV-2 infection was not associated with increased risk of current symptoms, impaired daily activities, reduced quality of life, hospitalization, ICU admission or death at follow-up - i.e. they do not get long-COVID. Therefore, it would not be appropriate to include them. This study is focused specifically on long-COVID following symptomatic infection.

The start date of the study is the same as the start date of PCR testing in Scotland. However, it is possible that someone had a previous positive PCR test, for example, conducted in another country. Also, the results of lateral flow tests, self-conducted at home, are not recorded on the register. Since, in these situations, we could not validate whether a previous test was positive (plus the test date) or whether it was mis-reporting, people who reported a previous positive test that was not recorded on the register were excluded from the study.

Are there data on severe covid-19, i.e. whether individuals were hospitalized at or around their test dates? Along this line, to whom do the findings apply? I think this needs to be made more clear in abstract, title etc.

Within Long-CISS, severe infection was defined as hospital admission for SARS-CoV-2 infection. Severity is included as a covariate in multivariate analysis.

We have added the crude prevalence of one or more ongoing symptom following severe and mild infection to the text. Unfortunately, the methodology used to calculate population attributable risk cannot be applied to mild and severe infections because all of them are infected and severity is a detailed version of the exposure variable, meaning that severity and test status are strongly correlated. This has been added to the discussion as a study limitation.

Reviewer #3 (Remarks to the Author):

Thank you for the opportunity to review this paper. Examining the symptom burden of long COVID using a matched cohort framework provides us a clearer understanding of the symptoms attributable to long COVID. As the authors point out, this approach has been examined in other populations and the addition of this large cohort is useful to get closer to understanding the prevalence of this condition. However, I have a few suggestions below that could benefit the reader. Many of my questions are around how the symptom onset and duration was captured, which is important to clarify when applying the WHO definition, as new onset and at least 2 months of duration are key components.

Clarifying Questions

1. Ln 154: "PCR tests took place between the 20th of April 2020 and the 31st of May 2022. Questionnaires were completed between the 10th of May 2021 and the 14th of November 2022." Given these dates, it seems possible that for some participants, the initial questionnaire was administered at least one year since the index COVID infection. Would these participants have only contributed data at the 12 month timepoint? Please clarify.

Yes, this is correct. Some participants were recruited at 12 months follow-up.

2. Ln 159: Please clarify how you determined the variant of concern for the uninfected matched group. From Hastie 2022 Nat Comm paper it appears this was the variant of concern at the time of study invitation.

No. The variant of concern is defined as the dominant variant ($\geq 95\%$ of infections) in the UK at the time of the index test (not at the time of recruitment). For some periods there is no dominant variant. This is at a population rather than individual level. Therefore, the period is defined in the same way for SARS-CoV-2 positive participants and the uninfected matched group. The Cog-UK Mutation Explorer website has been added to the manuscript.

3. Ln 189: "This is based on us applying the WHO definition of long-COVID as one or more persistent or new symptom."

a. How were the symptoms for both infected and matched controls determined to be new onset? Was it from Q9 on the questionnaire? If Q9 was utilized to determine new onset, then please comment on the possibility that exacerbation of an existing symptom prior to COVID infection was captured since the participants were asked "Are any of these new or worse since your Covid-19 test?"

The WHO definition includes current symptoms that are new and current symptoms that are persistent. Therefore, we did not need to differentiate new from persistent symptoms, rather we needed to differentiate current symptoms that were due to prior infection from those that would have occurred anyway; hence the need for a comparison group. Our previous publication investigates whether symptoms persist, resolve or occur as late new symptoms:

Hastie, C.E., Lowe, D.J., McAuley, A. et al. Natural history of long-COVID in a nationwide, population cohort study. Nat Commun 14, 3504 (2023). <https://doi.org/10.1038/s41467-023-39193-y>

b. The WHO definition emphasizes the minimum symptom duration of at least 2 months. How did you determine if the participant's reported symptoms lasted for at least this duration? From my reading of the questionnaire, it does seem possible that the patient may report a new symptom since COVID infection but the duration may not have been at least 2 months.

We accept this is a limitation of the study and have added it to the limitations section of the discussion.

4. Ln 213: "...with a positive PCR test was invited along with a comparison group who had had a negative test but never a positive test" Please comment on the availability of antigen tests for the cohort. Could someone with a positive antigen test be contained in the matched cohort group? Please comment on this potential misclassification if so.

Antigen tests were not available. A paragraph on the risk of misclassification bias had been added to the discussion.

5. Ln 217: "An SMS text prompted online questionnaire (Supplementary Figure 1), self-completed at six, 12 and 18 months following the index PCR test"

a. Could you comment on the response rate to the text questionnaire by infection status?

Under the data sharing agreement we have with the data custodian we are not provided with individual level data on e.g. infection status or response rate. We cannot stratify response rates at different stages of follow-up by infection status because we do not know if those who change infection status responded or not, furthermore we do not know which participants were invited more than once.

b. Can you describe further how SMS texts were sent out? From Hastie 2022 Nat Comm paper it appears the this was at health clinic level but also contact tracing databases.

No it was all performed by one single national system. The following sentence has been added to the methods "The National Health Service (NHS) Scotland platform that provided PCR result notifications identified eligible participants and invited them via automated SMS text messages."

6. Table 1: Can you further describe the time periods used for the variant periods, especially "no dominant"?

Periods were defined as having no dominant variant when no single variant accounted for $\geq 95\%$ of cases genotyped that week. This has been added to the methods. Further explanation is provided in response to question 2 above.

REVIEWER COMMENTS

Reviewer #2 (Remarks to the Author):

Thank you for clarifications. I am unsure whether this work has been much strengthened. In my first review I discussed a number of possible biases that I don't think have been sufficiently addressed. A general comment to the authors is to provide more structure in response to reviewers. Please include a thorough response to the specific comments, where reviewer concerns are met. Also include a detailed description of actions taken in the manuscript, including page and line numbers and marked edits, which should be copied into the response letter. In this way it will be a quick check for the reviewer to see whether concerns are met and how they are met, allowing the reviewer to quickly assess whether he or she thinks it is sufficient. As the response reads, it is unclear what are responses to me as a reviewer, what are the original parts and what are edits and improvements. This unclarity will cover several of my comments.

1. Authors response "Furthermore, selection bias may be present in those who were tested for SARS-CoV-2, those who completed the questionnaire, and those who consented to linkage." This response to reviewer and action in manuscript regarding possible selection bias is not sufficient. Please describe how each of these sources of selection bias could arise and how it might impact findings (over or underestimation) and how the issue of selection bias makes or does not make findings generalizable, i.e. to whom do your findings apply.

2. The authors have added that individuals were included with (first positive test or, for the comparison group, most recent negative test), indicating there are systematic differences in selection based on testing. A thorough discussion is needed as to how this might impact findings. If no data exist, the authors should elaborate on test patterns of individuals with a positive test compared to individuals with a range of negative tests. There might differences based on healthcare behaviour but also differences in testing patterns induced. For example, some occupational groups of activities in society required a negative test prior to certain time points in certain periods of pandemic etc.

3. The response to reviewer and edits to manuscript are not consistent with regards to time units for matching. Further, I mentioned possible time-based biases however there is no discussion of how it might have impacted on findings? "Those who tested positive for SARS-CoV-2 (n=3,065) were matched by month of test, age, sex, and geographical region to adolescents who tested negative (n=3,739)."

4. In my initial review, I pointed out that the aim of the study is not fully answered by the analyses performed. The response from reviewer is unclear and I cannot see how my concerns are met. Were revisions made, and where can they be found in the manuscript? Why, or why not, am I wrong and how did the authors meet my concerns? Please clarify. For example, "Figure 1 presents subgroup estimates at 6-month follow-up because this is the largest sample." Why is it good to only have subgroup estimates for the largest sample? And why only explain this to me as a reviewer and not to future readers?

5. Definition of long covid. Again, my initial concerns are not addressed by the authors. It is unclear whether any revisions have been made and what are the original parts of the manuscript.

6. Author response, but not in paper: "Omitting people infected early in the pandemic would be inappropriate as we would lose the people with the longest follow-up and the results would not be representative of the pandemic as a whole." The authors is correct about the length of follow-up, but not in the representativity of the pandemic as a whole. The entire pandemic is not representative of itself – it consists of different phases with different variants, different vaccination statuses and different assumed disease severity. The paper does not reflect or discuss this.

Reviewer #3 (Remarks to the Author):

Hastie and colleagues estimate the prevalence of long COVID in a Scottish population and adds to the body of literature describing the burden of long COVID. The authors clarified several points related to possible selection bias which were of concern to multiple reviewers. Thank you for your clarifications which have made this a much stronger paper. I would recommend to the authors to change the wording of this line: "This is based on us applying the WHO definition of long-COVID as

one or more persistent or new symptom” to make it clear that a modified WHO definition was applied to the cohort, given that the authors could not measure symptom duration of at least 2 months as required by the WHO definition. The use of a 4 week or 2 month duration to define long COVID could have significant implications on the count of long COVID cases and it would be clearer for the reader to clarify the symptom duration required. Thank you for your work and contribution to the field.

REVIEWER COMMENTS RECEIVED 9TH OF OCTOBER 2023

Reviewer #2 (Remarks to the Author):

Thank you for clarifications. I am unsure whether this work has been much strengthened. In my first review I discussed a number of possible biases that I don't think have been sufficiently addressed. A general comment to the authors is to provide more structure in response to reviewers. Please include a thorough response to the specific comments, where reviewer concerns are met. Also include a detailed description of actions taken in the manuscript, including page and line numbers and marked edits, which should be copied into the response letter. In this way it will be a quick check for the reviewer to see whether concerns are met and how they are met, allowing the reviewer to quickly assess whether he or she thinks it is sufficient. As the response reads, it is unclear what are responses to me as a reviewer, what are the original parts and what are edits and improvements. This unclarity will cover several of my comments.

In the manuscript all amendments are marked with tracked changes.

The previous reviewer comments and responses are appended below. Where changes were made to the text, the page and line numbers are now provided as tracked changes.

1. Authors response "Furthermore, selection bias may be present in those who were tested for SARS-CoV-2, those who completed the questionnaire, and those who consented to linkage." This response to reviewer and action in manuscript regarding possible selection bias is not sufficient. Please describe how each of these sources of selection bias could arise and how it might impact findings (over or underestimation) and how the issue of selection bias makes or does not make findings generalizable, i.e. to whom do your findings apply.

The following has been added to the discussion: "During the time period when index PCR tests were conducted testing was available to everyone free of charge. However, people might be less likely to have been tested if their symptoms were mild resulting in some bias in testing. Furthermore, selection bias in questionnaire completion could potentially lead to overestimation of associations if having ongoing symptoms made participation more likely, or alternatively underestimation of associations if having more severe ongoing symptoms affected the ability to participate. In terms of linkage consent it is difficult to determine what direction of effect this might have. Despite this limitation our methodology represents a pragmatic recruitment method that allows representative response at a population level." (page 8 line 23 to page 9 line 7)

2. The authors have added that individuals were included with (first positive test or, for the comparison group, most recent negative test), indicating there are systematic differences in selection based on testing. A thorough discussion is needed as to how this might impact findings. If no data exist, the authors should elaborate on test patterns of individuals with a positive test compared to individuals with a range of negative tests. There might differences based on healthcare behaviour but also differences in testing patterns induced. For example, some occupational groups of activities in society required a negative test prior to certain time points in certain periods of pandemic etc.

As part of the governance agreement with the data providers, we were only provided with the classification of participants (positive/negative) and the index date. We were not provided with the dates and results of repeated tests in the same individual. Participants were invited based on their first positive test because this is the first, or only, time they were infected with SARS-CoV-2 infection and therefore the first time point at which they became at risk of developing long-COVID. People who only ever tested negative were invited based on their most recent negative test because people who had negative tests often went on to have positive tests later in the pandemic and therefore we needed to ensure that they had never been infected (as far as we know) at the index date.

3. The response to reviewer and edits to manuscript are not consistent with regards to time units for matching. Further, I mentioned possible time-based biases however there is no discussion of how it might have impacted on findings? “Those who tested positive for SARS-CoV-2 (n=3,065) were matched by month of test, age, sex, and geographical region to adolescents who tested negative (n=3,739).”

There is no inconsistency in our reporting of time units for matching. Participants who tested positive and negative were matched in units of three-month periods. The sentence quoted above is incorrectly attributed to our study. It is in fact a description of a study by Stephenson et al. (2022) in the discussion.

4. In my initial review, I pointed out that the aim of the study is not fully answered by the analyses performed. The response from reviewer is unclear and I cannot see how my concerns are met. Were revisions made, and where can they be found in the manuscript? Why, or why not, am I wrong and how did the authors meet my concerns? Please clarify. For example, “Figure 1 presents subgroup estimates at 6-month follow-up because this is the largest sample.” Why is it good to only have subgroup estimates for the largest sample? And why only explain this to me as a reviewer and not to future readers?

The referee is incorrect in stating that our manuscript “Only has sub-group estimates at 6-month follow-up”. Supplementary Table 1 contains the subgroup estimates at all follow-up times. Figure 1 (in the main text) presents the results for only one follow-up time point (6 months) for simplicity. It would be overcrowded if we presented all the data and this is unnecessary since the full set of results is contained in the Supplementary Table. 6-months was chosen for Figure 1 since it is by far the largest sample and therefore the calculated estimates of true prevalence are more precise (narrower confidence intervals). Readers understand the concept that larger sample sizes increase statistical power and precision, and they can view the full results in the Supplementary Table.

5. Definition of long covid. Again, my initial concerns are not addressed by the authors. It is unclear whether any revisions have been made and what are the original parts of the manuscript.

Our primary outcome is long-Covid, defined as one or more self-reported symptom at follow-up. The text has been amended to clarify this. (page 11 lines 5-6)

6. Author response, but not in paper: “Omitting people infected early in the pandemic would be inappropriate as we would lose the people with the longest follow-up and the results would not be representative of the pandemic as a whole.” The authors is correct about the length of follow-up, but not in the representativity of the pandemic as a whole. The entire pandemic is not representative of itself – it consists of different phases with different variants, different vaccination statuses and different assumed disease severity. The paper does not reflect or discuss this.

We disagree with the reviewer. Generalisability is dependent on representativeness. Representativeness is defined as the extent to which the study sample is representative of the target population. As stated in the manuscript title the aim of the study is to determine the true prevalence of long-COVID in the general population of a whole nation. The aim is not to determine the prevalence of long-COVID among people infected with a particular variant. Therefore, our target population is everyone infected by SARS-CoV-2 in Scotland and our study sample reflects this. The statement that “the entire population is not representative of itself” is fundamentally untrue since there is no study sample more representative than a 100% sample. The reviewer is confusing representativeness (which dictates that a study sample must include variability where it exists in the target population) with homogeneity (the use of selection to minimise variability - a process used for example in randomised controlled trials to maximise statistical power but at the expense of representativeness and generalisability).

Regarding “the paper does not reflect or discuss this” - Our manuscript does report and adjust for factors that varied over time including: dominant variant, vaccination status at the time of infection, and disease severity. (Table 1 pages 13-14, Figure 1 page 17, Supplementary Table 1)

Reviewer #3 (Remarks to the Author):

Hastie and colleagues estimate the prevalence of long COVID in a Scottish population and adds to the body of literature describing the burden of long COVID. The authors clarified several points related to possible selection bias which were of concern to multiple reviewers. Thank you for your clarifications which have made this a much stronger paper. I would recommend to the authors to change the wording of this line: “This is based on us applying the WHO definition of long-COVID as one or more persistent or new symptom” to make it clear that a modified WHO definition was applied to the cohort, given that the authors could not measure symptom duration of at least 2 months as required by the WHO definition. The use of a 4 week or 2 month duration to define long COVID could have significant implications on the count of long COVID cases and it would be clearer for the reader to clarify the symptom duration required. Thank you for your work and contribution to the field.

Thank you. We have amended the wording as requested. (page 7 lines 11-16)

REVIEWER COMMENTS RECEIVED 23RD OF MAY 2023 – ADDITIONAL RESPONSE INFORMATION ADDED (SHOWN AS TRACKED CHANGES) TO INDICATE WHERE THE MANUSCRIPT TEXT HAS BEEN AMENDED FOLLOWING FURTHER COMMENTS RECEIVED 9TH OF OCTOBER 2023

Reviewer #1 (Remarks to the Author):

Hastie and colleagues determined the ‘true prevalence’ of Long Covid in the general Scottish population using a largescale nationwide study. They found 6, 12, and 18 months following symptomatic infection 65%, 68% and 73% of people reported no, or incomplete, recovery respectively. Of note, the above symptoms were also commonly reported among test negative individuals at 6 (51%), 12 (55%) and 18 months (56%). This demonstrates the relative commonness of these symptoms in the population at any given time and subsequently the importance of a test-negative control group.

This ambidirectional, general population cohort study included people who had symptomatic laboratory-confirmed SARS-CoV-2 infections and a test negative comparison group, matched for age, sex, and deprivation, allowing novel insights into Long Covid that have not yet been extensively reported. On page 6, the authors make reference to their previous analysis of the same cohort. I think it would be helpful to the reader for the authors to provide clarity on how this work differs from and/or expands on that previously published in Nature Communications in October 2022.

The following sentence has been added “This work expands on that previously published by including additional waves of questionnaires and focusing analysis on the calculation of the prevalence of ongoing symptoms attributable to SARS-CoV-2 infection.” (page 4 lines 23-25)

The methods used to collect the data were valid, and the authors have provided a copy of self-reported symptom questionnaire. The manuscript is very well written; clear, precise, and easy to understand. The included figures and tables visually present complex data that is easy to interpret. The authors state whilst 64.5%-72.6% of people report at least one symptom 6-18 months following SARS-CoV-2 infection, only 6.6%-10.3% are likely to have long-COVID. They suggest the symptoms of the remainder are likely to have occurred without SARS-CoV-2 infection, but some people may mistakenly attribute them to long-COVID. Further explanation as to why this occurred and recommendations on how to avoid this misclassification during future research would be beneficial.

The following sentence has been added “Further work is required to refine the definition and diagnosis of long-COVID and support appropriate management.” (page 6 lines 20-21)

The authors conclude when taking into account the prevalence of Long Covid symptoms in non-infected matched controls and adjusting for potential confounders the prevalence of Long Covid is lower than previously reported in the literature (10.3% at 18 months). Though the authors make reference to the work by Ballering and colleagues I suggest the also include the work by Stephenson et al., 2022 ([https://www.thelancet.com/journals/lanchi/article/PIIS2352-4642\(22\)00022-0/fulltext](https://www.thelancet.com/journals/lanchi/article/PIIS2352-4642(22)00022-0/fulltext)), that also included a test negative control group. Since the authors state the attributable prevalence was higher in those who had had received more vaccination doses prior to infection I encourage them to compare their findings to those by Antonelli and colleagues (<https://pubmed.ncbi.nlm.nih.gov/34480857/>).

Stephenson et al. (2022) (page 6 line 24) and Antonelli et al. (2022) (page 8 line 8) have been added to the discussion.

Reviewer #2 (Remarks to the Author):

In the study “True prevalence of long-COVID in a nationwide, population cohort study” the authors aim to determine the prevalence of symptoms attributable to SARS-CoV-2 infection, taking account of background rates and confounding, in a nationwide population cohort study of 198,096 Scottish adults. The authors conclude that long covid symptoms cannot be linked to previous SARS-CoV-2 infection. The study has an interesting topic and results and could potentially add to the existing literature; however I have several concerns regarding the methods used and interpretation. Given many unclarities and limitations, I am not sure about the validity of findings and thus the significance to the field.

Major:

Selection into the study may greatly influence results. Several studies have shown that individuals testing negative, which make up the control group in this study, may be particularly health-conscious individuals who test themselves more often and also generally report more health complaints. Thus, the claim of no difference in prevalence for individuals testing positive vs negative may be explained by selection bias. Please additionally include a control group of individuals who were non-tested. If not available, severe selection bias needs to be discussed including how it impacted on the estimates (from all sources, i.e. both from the choice of control group, from lack of questionnaire reporting and from non-consent to linkage).

Potential participants were invited using contact details held by NHS Scotland in their PCR result database used for the SARS-CoV-2 test and protect initiative. Therefore, it is not possible to include a control group who were not tested.

People who did not get tested were also subject to potential bias and were more prone to undiagnosed SARS-CoV-2 infection.

The possibility of selection bias has been added to the discussion. (page 8 lines 22-23)

Testing procedures in the community/study sample and how data on testing was managed in the study should be clearly described and their impact on results should be thoroughly discussed. Please provide data on how many tests were performed per individual and which test was selected in case of multiple tests. Most individuals have multiple negative and positive tests, and which test was chosen might impact on findings. Please describe and justify your approach.

We do not know how many tests were performed per individual because we do not have access to these individual level data. The index test was the first positive test or, for the comparison group, most recent negative test. This information has been added to the methods. (page 10 line 12)

For example, participants were included with an index PCR test and individuals testing negative were reallocated to the infected group if and when they tested positive. How was this handled in the study design and methods? If, for example, an individual was included in the negative group and reported their

symptoms at six months, before testing positive at 8 months. Would he or she be in the positive group at 12 months? Wouldn't this method require a prospective design, which this study doesn't have? This is important to describe given that relatively few were infected early on during the pandemic and many were infected later.

Yes, this individual would be in the positive group at 12 months with length of follow-up adjusted according to the date of their positive test (i.e. the clock restarts when they become positive). Their follow-up as an initial negative participant was censored at the time they become positive. Long-CISS employs an ambidirectional study design – that is, it includes both retrospective and prospective components. This information has been added to the methods. (page 10 line 1)

Along this line, the design of the study is unclear. The authors name it a “population cohort study”. Please describe under what circumstances this is a population cohort study. Is it prospective or retrospective? Why is this not a case control study – I think, from the descriptions, that cases were selected based on testing positive and controls testing negative were identified, retrospectively? Yet, it is not necessarily a case-control study, the methods including matching are too poorly described for the reader to judge. In the evidence hierarchy, case control studies using retrospective methods provide less strong evidence, for example on cause and effect, than prospective cohort studies. Prospective cohort studies on the topic do exist and should be referred to (example Magnusson et al., Scientific Reports, 2023). Given the design, the authors need to present descriptions of how individuals who could not continue their participation in the study were handled. Did they contribute in the study until they e.g. moved or died, and were censored from that time, or did you require complete followup? The latter would introduce further selection bias. If the authors think they have a prospective design, please use methods accordingly.

The study is a population cohort study. The population is the Scottish general population from which participants were recruited. If they were recruited from e.g. hospital attendance, it would have been a hospital cohort. The study is an ambidirectional cohort study. An ambidirectional cohort combines both retrospective and prospective cohort designs. It recruits historical cases and adds new incident cases prospectively. It is not a case control study. A case-control study classifies and recruits people based on the outcomes of interest. Therefore, a case control study would recruit people with long-COVID (cases) and people without long-COVID (controls), then ascertain their exposure of interest. In contrast, a cohort study classifies and recruits people based on their exposure of interest. Therefore, it recruits people with SARS-CoV-2 infection (exposed) and people without SARS-CoV-2 infection (not exposed), then ascertains their outcome of interest. Cohort studies are a stronger epidemiological design largely because case-control studies are prone to survival bias. Both cohort and case-control studies provide evidence of association that should not be assumed to be causal. Temporality is one of the criteria that strengthens the likelihood of an association being causal. In our study the test date precedes the questionnaire date by at least 6 months. Analysis is based on completed questionnaires at follow-up. Therefore, people who do not complete questionnaires (due to death or other reason) are not included (in either the numerator or denominator). We have added a note that people dying from long-COVID over time could contribute to a fall in the prevalence of long-COVID over follow-up. (page 8 lines 18-20)

Reference to Magnusson et al. has been added to the manuscript. (page 7 line 23)

The matching needs a more thorough description. For example, time period was matched on, and from Table 1, it looks like the time unit included was based on dominant variant, typically lasting for half a year to a whole year. Time based biases might be introduced due to long durations of time matching units. Please describe and consider if time based biases can be minimized by design and methods. Could you match on calendar month or even week?

Three-month periods were used to group potential participants and match positive and negative tests. This information has been added to the methods. (page 10 line 4) The time units based on dominant variant were only used to adjust multivariate analyses.

The aim of this study is not fully answered by the analyses performed. For example, Figure 1 only covers adjusted attributable prevalence at 6 months, not 12 or 18 months. And Table 1 includes many symptoms but is not broken down on subgroups. Please consider skipping one of these parts and perform a more thorough analyses of one of the sub-aims, i.e. focus either on all the symptoms and skip the subgroups, or focus on all the subgroups and provide one outcome.

Overall adjusted attributable prevalence estimates are provided for each follow-up time in the text. Figure 1 presents subgroup estimates at 6-month follow-up because this is the largest sample.

The study needs a clear definition of what is meant by “long covid” and what are the clinical implications of the findings, given all the limitations particularly regarding selection bias and possible confounding. It is unclear what is being measured and of main interest in this study. “The outcome was one or more self-reported symptom at follow-up.” However, the results section includes prevalence of a wide range of symptoms, which is inconsistent with methods and also inconsistent with the title. Do the authors measure long covid symptoms, or long covid?

Our primary outcome is long-Covid, defined as one or more self-reported symptom at follow-up. This is determined from the response to a long tick list of symptoms experienced within the last week. Table 2 is included to provide information on the individual symptoms reported and their prevalence.

The authors also need a better presentation of their data to support their conclusion and several claims in the discussion section. Confidence intervals of prevalence estimates are lacking, i.e. one cannot say that 64.5% is higher than 50.8% if CIs are overlapping which we do not know whether they are (and similar for the symptom reporting at 6, 12 months etc). Please provide CIs of all prevalence estimates, both crude and adjusted. For the adjusted estimates, please provide the prevalence with their CI, not only the adjusted difference with their CIs. This is important for the transparency of this study. It is always important to show the uncertainty around all estimates.

We have added confidence intervals around the crude attributable prevalences. (page 5 lines 19-20) Confidence intervals for the adjusted attributable prevalences were already included in the Figure but have now been added to the text. (page 5 lines 21-22) We have not added confidence intervals around the actual prevalences in either group as these are not estimates.

Minor:

From table 2, why do symptom reporting seem to increase from six months to 18 months? In previous

research, complaints seem to decrease over time (see e.g. Skyrud et al., PlosOne, 2021). Again, confidence intervals are needed to determine if there is a real increase. Reasons for the potential increase, both for case and control group need to be discussed. Please consider running a within- and between individual model to take within-person clustering of symptoms over time into account.

There are a lot of potential explanations for changes by stage of follow-up. In the infected group it may be due to a real increase in long-COVID due to re-emergence or new occurrence of symptoms or be due to retention bias whereby retention is higher in those with symptoms. Both groups gain 6 months of age between questionnaires and most symptoms increase with age. Both SARS-CoV-2 infection and many of the symptoms reported at follow-up vary by season. However, this is more likely to explain differences between 6 and 12 months follow-up and between 12 and 18 months (not between 6 and 18 months which would occur during the same season). We have added this to the limitations section. (page 8 lines 13-17) We have already reported between individual results. Reporting within individual changes would markedly reduce statistical power. We report confidence intervals for derived estimates (crude and adjusted attributable percentages) but not for absolute prevalence as per convention.

The discussion section lacks important references and should include also what we know about prevalence from registerbased research. See for example omicron vs delta, Magnusson et al, Nat Comm 2022. Please also discuss the generalizability of findings and how self-reported data may be prone to bias and lack of generalizability. Along this line, selection bias is particularly important, for example that only 89% consented to linkage. And, very important, please provide data on the source population. How many were invited to respond to questionnaires and how many responded, at each measurement time during follow-up? What were the characteristics of non-responders and how might it impact on findings that these did not respond? As now, we only find the numbers and characteristics of responders yet we do not know what characterized non-responders or how many they were (participated at all visits during followup etc? Please see comment above on design).

We have limited, aggregated data on non-responders. The following has been added to the text "Compared with those who did not provide consent, participants in the final sample were more likely to be female (58.8% vs 51.8%; p-value <0.001), were older (>40 years 64.0% vs 51.1%; p-value <0.001) and slightly more deprived (most deprived SIMD quintile 20.8% vs 20.4%; p-value <0.001)." (page 5 lines 8-11)

It is controversial to include ethnicity in scientific research. Please focus on socioeconomic or residential region as markers of background.

We disagree completely with this statement. Whilst ethnicity and socioeconomic status may be correlated to some extent, they are completely different concepts. We have data on both and have included both. Previous studies have provided consistent evidence of differences in infection rates and outcomes by ethnic group, independent of socioeconomic status. Therefore, to ignore this as a potential confounder would be inappropriate.

Tables and figures should be self-explanatory. Please provide descriptions of abbreviations and measurements in legends.

Where missing, this information has been added to the manuscript.

Table 1 and Supplementary Table 1 contain massive amounts of data and is better presented in figure format, with confidence intervals.

.....

We disagree. Table 1 is summary data and, as per convention, is presented as a Table. We believe both would be messy, bigger and harder to interpret as Figures.

I don't understand the data on variant, please describe pre VOC, no dominant 1, 2, etc

The variant of concern is defined as the dominant variant ($\geq 95\%$ of infections) in the UK at the time of index test. Periods were defined as having no dominant variant when no single variant accounted for $\geq 95\%$ of cases genotyped that week. This has been added to the methods, as has the Cog-UK Mutation Explorer website. (page 10 lines 24-25) Pre-VOC is the period early in the pandemic prior to any new, dominant variants of the original pathogen being identified.

"Pre-existing health conditions were ascertained from self-report using the questionnaire, as well as linkage to previous hospitalizations and dispensed prescriptions." Please describe this linkage, how and when was it performed, what type of hospitalizations were included and for what period. How were the data compiled?

The following has been added to the methods "Additional data were obtained through linkage to electronic health records - both five years prior to their index test and subsequent to the test (up to January 2022) - on hospitalizations (Scottish Morbidity Record 01/04), dispensed prescriptions (Prescribing Information System), vaccinations, and death certificates (General Registrar Office)." (page 10 lines 18-21)

I can see the authors included the initial phase of the pandemic, from April 2020, when there was a lot of fear and anxiety. At the time, long covid was not known or established and symptom reporting may be increased just by the knowledge that one had the initial variant. Please include in discussion and consider omitting the first half year of testing in a sensitivity analyses. Also, to shed light on the generalizability of findings, please provide a timeline showing the number of tests from April 2020 to May 2022 as well as the proportion of all the tests being positive.

We did not ask people if they have long-COVID. We asked if they have any of a long list of symptoms. We then defined long-COVID based on their responses. We have already included in the analysis the phase of the pandemic (using VOC data). Omitting people infected early in the pandemic would be inappropriate as we would lose the people with the longest follow-up and the results would not be representative of the pandemic as a whole.

The COVID-19 & Respiratory Surveillance in Scotland Dashboard (<https://scotland.shinyapps.io/phs-respiratory-covid-19/>) provides information on testing and positivity rates over time. This has been added to the methods section. (page 10 lines 8-10)

Why were people with asymptomatic infection excluded and how might it impact on findings? Again, please describe test criteria as one might wonder why asymptomatic individuals would be tested? Detail

on methods with justification that are lacking: What were the criteria for being tested? Where and how did the testing take place? Was there routine testing, e.g. with hospital admission? How were home testing managed? Did the test criteria change during the inclusion period and were they equal for everyone included? Why were some previous positive tests not recorded on the database (and how might it impact on results)? Etc.

PCR testing was freely available to the population for most of the period covering index tests. The reason for the test being conducted is not recorded. However, during the pandemic certain occupational groups (hospital and social care workers) were routinely tested as were people travelling between countries. This resulted in the detection of asymptomatic infections. Our previous publication (Hastie, C.E., et al. Outcomes among confirmed cases and a matched comparison group in the Long-COVID in Scotland study. Nature Communications 13, 5663 (2022)) found that asymptomatic SARS-CoV-2 infection was not associated with increased risk of current symptoms, impaired daily activities, reduced quality of life, hospitalization, ICU admission or death at follow-up - i.e. they do not get long-COVID. Therefore, it would not be appropriate to include them. This study is focused specifically on long-COVID following symptomatic infection.

The start date of the study is the same as the start date of PCR testing in Scotland. However, it is possible that someone had a previous positive PCR test, for example, conducted in another country. Also, the results of lateral flow tests, self-conducted at home, are not recorded on the register. Since, in these situations, we could not validate whether a previous test was positive (plus the test date) or whether it was mis-reporting, people who reported a previous positive test that was not recorded on the register were excluded from the study.

Are there data on severe covid-19, i.e. whether individuals were hospitalized at or around their test dates? Along this line, to whom do the findings apply? I think this needs to be made more clear in abstract, title etc.

Within Long-CISS, severe infection was defined as hospital admission for SARS-CoV-2 infection. Severity is included as a covariate in multivariate analysis.

We have added the crude prevalence of one or more ongoing symptom following severe and mild infection to the text. (page 6 lines 5-8) Unfortunately, the methodology used to calculate population attributable risk cannot be applied to mild and severe infections because all of them are infected and severity is a detailed version of the exposure variable, meaning that severity and test status are strongly correlated. This has been added to the discussion as a study limitation. (page 9 lines 9-14)

Reviewer #3 (Remarks to the Author):

Thank you for the opportunity to review this paper. Examining the symptom burden of long COVID using a matched cohort framework provides us a clearer understanding of the symptoms attributable to long COVID. As the authors point out, this approach has been examined in other populations and the addition of this large cohort is useful to get closer to understanding the prevalence of this condition. However, I have a few suggestions below that could benefit the reader. Many of my questions are around how the symptom onset and duration was captured, which is important to clarify when applying the WHO

definition, as new onset and at least 2 months of duration are key components.

Clarifying Questions

1. Ln 154: "PCR tests took place between the 20th of April 2020 and the 31st of May 2022. Questionnaires were completed between the 10th of May 2021 and the 14th of November 2022." Given these dates, it seems possible that for some participants, the initial questionnaire was administered at least one year since the index COVID infection. Would these participants have only contributed data at the 12 month timepoint? Please clarify.

Yes, this is correct. Some participants were recruited at 12 months follow-up.

2. Ln 159: Please clarify how you determined the variant of concern for the uninfected matched group. From Hastie 2022 Nat Comm paper it appears this was the variant of concern at the time of study invitation.

No. The variant of concern is defined as the dominant variant ($\geq 95\%$ of infections) in the UK at the time of the index test (not at the time of recruitment). For some periods there is no dominant variant. This is at a population rather than individual level. Therefore, the period is defined in the same way for SARS-CoV-2 positive participants and the uninfected matched group. The Cog-UK Mutation Explorer website has been added to the manuscript. (page 10 line 24)

3. Ln 189: "This is based on us applying the WHO definition of long-COVID as one or more persistent or new symptom."

a. How were the symptoms for both infected and matched controls determined to be new onset? Was it from Q9 on the questionnaire? If Q9 was utilized to determine new onset, then please comment on the possibility that exacerbation of an existing symptom prior to COVID infection was captured since the participants were asked "Are any of these new or worse since your Covid-19 test?".

The WHO definition includes current symptoms that are new and current symptoms that are persistent. Therefore, we did not need to differentiate new from persistent symptoms, rather we needed to differentiate current symptoms that were due to prior infection from those that would have occurred anyway; hence the need for a comparison group. Our previous publication investigates whether symptoms persist, resolve or occur as late new symptoms:

Hastie, C.E., Lowe, D.J., McAuley, A. et al. Natural history of long-COVID in a nationwide, population cohort study. Nat Commun 14, 3504 (2023). <https://doi.org/10.1038/s41467-023-39193-y>

b. The WHO definition emphasizes the minimum symptom duration of at least 2 months. How did you determine if the participant's reported symptoms lasted for at least this duration? From my reading of the questionnaire, it does seem possible that the patient may report a new symptom since COVID infection but the duration may not have been at least 2 months.

We accept this is a limitation of the study and have added it to the limitations section of the discussion. (page 7 lines 14-16)

4. Ln 213: "...with a positive PCR test was invited along with a comparison group who had had a negative

test but never a positive test" Please comment on the availability of antigen tests for the cohort. Could someone with a positive antigen test be contained in the matched cohort group? Please comment on this potential misclassification if so.

Antigen tests were not available. A paragraph on the risk of misclassification bias had been added to the discussion. (page 9 lines 16-21)

5. Ln 217: "An SMS text prompted online questionnaire (Supplementary Figure 1), self-completed at six, 12 and 18 months following the index PCR test"

a. Could you comment on the response rate to the text questionnaire by infection status?

Under the data sharing agreement we have with the data custodian we are not provided with individual level data on e.g. infection status or response rate. We cannot stratify response rates at different stages of follow-up by infection status because we do not know if those who change infection status responded or not, furthermore we do not know which participants were invited more than once.

b. Can you describe further how SMS texts were sent out? From Hastie 2022 Nat Comm paper it appears the this was at health clinic level but also contact tracing databases.

No it was all performed by one single national system. The following sentence has been added to the methods "The National Health Service (NHS) Scotland platform that provided PCR result notifications identified eligible participants and invited them via automated SMS text messages." (page 10 lines 6-8)

6. Table 1: Can you further describe the time periods used for the variant periods, especially "no dominant"?

Periods were defined as having no dominant variant when no single variant accounted for $\geq 95\%$ of cases genotyped that week. This has been added to the methods. (page 10 lines 24-25) Further explanation is provided in response to question 2 above.